# Atomic-scale manipulation of single-polaron in a two-dimensional semiconductor

Huiru Liu [1,2,9], Aolei Wang [3,9], Ping Zhang[1,2], Chen Ma[1,2], Caiyun Chen[1,2], Zijia Liu[1,2,4], Yi-Qi Zhang[1,2], Baojie Feng [1,2,5], Peng Cheng [1,2], Jin Zhao [3,6,7,8] ✉, Lan Chen [1,2,4] ✉ & Kehui Wu [1,2,4,5] ✉

Polaron is a composite quasiparticle derived from an excess carrier trapped by local lattice distortion, and it has been studied extensively for decades both theoretically and experimentally. However, atomic-scale creation and manipulation of single-polarons in real space have still not been achieved so far, which precludes the atomistic understanding of the properties of polarons as well as their applications. Herein, using scanning tunneling microscopy, we succeeded to create single polarons in a monolayer two-dimensional (2D) semiconductor, $CoCl_2$. Combined with first-principles calculations, two stable polaron configurations, centered at atop and hollow sites, respectively, have been revealed. Remarkably, a series of manipulation progresses − from creation, erasure, to transition − can be accurately implemented on individual polarons. Our results pave the way to understand the physics of polaron at atomic level, and the easy control of single polarons in 2D semiconductor may open the door to 2D polaronics including the data storage.

Polaron is a fundamental physical phenomenon associated with the behavior of charges in insulators or semiconductors. Different from metal where the excess charge can flow as current and easily be compensated, insulators or semiconductors usually have charges stuck inside. For example, rubbing glass with fur can build up significant charges in both materials. Obviously, understanding the atomistic details of these charge states in insulators or semiconductors is fundamental. Excess charge (electron or hole) trapped by self-induced lattice distortion due to electron-phonon coupling interaction (EPI) is named as polaron[1]. Depending on the strength of EPI and the range of lattice distortion, polarons are usually divided into two categories: large polaron and small polaron. The large polaron is the result of weak long-range EPI (Frohlich model[2,3]), showing large polaron radius and metallic-like transport behavior of dressed carrier. In contrast, the small polaron is driven by short-rang

EPI (Holstein model[4,5]) or strong long-range EPI, and has a lattice distortion range comparable to the lattice constant and thermally activated hopping properties. Recently, polarons have been identified in a large variety of materials, such as alkali chloride KCl[6], transition metal oxides $TiO_2$[7–9], metal halide perovskites[10,11], organic semiconductors[12], blue phosphorene[13] and at the interfaces of metal dichalcogenides/STO[14].

Polaron has significant influences on the properties of semiconducting materials, such as lattice reconstructions[15], carrier mobility[16,17], surface adsorption and catalysis[18–20], ferromagnetic transition[21–23], superconductivity[24–26] and other many-body correlation states[27–29]. The investigations and applications of polaron have has been developed as an emerging field in nanotechnology, named as 'Polaronics'. So far, polarons have been experimentally accessed by various techniques such as Raman[30,31], THz

[1]Institute of Physics, Chinese Academy of Sciences, 100190 Beijing, China. [2]School of Physical Sciences, University of Chinese Academy of Sciences, 100190 Beijing, China. [3]Department of Physics, University of Science and Technology of China, 230026 Hefei, Anhui, China. [4]Songshan Lake Materials Laboratory, Dongguan, 523808 Guangdong, China. [5]Interdisciplinary Institute of Light-Element Quantum Materials and Research Center for Light-Element Advanced Materials, Peking University, 100871 Beijing, China. [6]ICQD/Hefei National Research Center for Physical Sciences at the Microscale, University of Science and Technology of China, 230026 Hefei, Anhui, China. [7]Department of Physics and Astronomy, University of Pittsburgh, Pittsburgh 15260 PA, USA. [8]Hefei National Laboratory, University of Science and Technology of China, 230088 Hefei, Anhui, China. [9]These authors contributed equally: Huiru Liu, Aolei Wang. ✉e-mail: zhaojin@ustc.edu.cn; lchen@iphy.ac.cn; khwu@iphy.ac.cn

spectroscopy[32], angle-resolved photoelectron emission spectroscopy[33,34], electron paramagnetic resonance[35], and scanning tunneling microscopy (STM)[9,15,36]. However, the capability to view, create and manipulate single polarons in real space still remains elusive, which limits the understanding and application of polaronics at atomic scale. Particularly, in previous studies defects or dopants have often been introduced into the system in order to create excess charge in the material, which inevitably resulted in polarons bonded with defects, and hindered the observation of intrinsic properties of polarons. A clean method to study pure polarons is thus highly desirable.

STM is an ideal tool for probing and manipulating local objectives with atomic precision[36]. In this work, we study single polarons in transition metal dichloride CoCl$_2$, an intrinsic magnetic 2D semiconductor, by STM. Through applying a voltage pulse, we can inject an electron into the monolayer CoCl$_2$ and create a single electron polaron. It is found that the polarons have two stable forms, labeled as type-I and type-II polarons. First principles calculations indicate that they can be described as spin-polarized electrons bonded to lattice distortions centered at either atop site or hollow site of the surface, respectively. Both types of polarons falls into the small polaron category[37], and are single spin information carriers. Moreover, the creation and erasure of single polarons, as well as transition between type-I and type-II polarons, can be realized by regulating the voltage pulse. Our work provides an ideal models system to study of intrinsic polaronics based on 2D materials.

## Results and discussion
### Growth and characterization of CoCl$_2$ monolayer
CoCl$_2$ is a layered material stabilized by weak interlayer van der Waals (vdW) interaction[38]. The atomic structure of CoCl$_2$ monolayer is shown in Fig. 1d. Each CoCl$_2$ monolayer consists of a sub-layer of Co atoms sandwiched between two Cl sub-layers. In our experiment, the CoCl$_2$ monolayer was grown on highly oriented pyrolytic graphite (HOPG) substrate. The STM image in Fig. 1a reveals that the monolayer CoCl$_2$ grows in a dendritic morphology starting from step edges of HOPG. The height of monolayer CoCl$_2$ is $480 \pm 5$ pm, clearly larger than that of HOPG ($340 \pm 5$ pm), as illustrated in Fig. 1a. Such dendritic growth mode can be described in the diffusion limited aggregation framework[39]. The atomic-resolution STM images of the monolayer CoCl$_2$ surface shown in Fig. 1b, c reveal a triangular lattice with periodicity of $354 \pm 2$ pm, consistent with the lattice parameters (354.5–355.3 pm) of CoCl$_2$ single crystal[40,41] and powder[42]. Moreover, depending on the crystallographic orientation of the CoCl$_2$ monolayer with respect to the HOPG substrate, different moiré patterns can be found on the surface of CoCl$_2$ monolayer. For examples, Fig. 1c exhibits a moiré periodicity of about 1.18 nm, whereas there is no moiré modulation in Fig. 1b. More domain boundaries are also shown in Supplementary Fig. 14.

The low temperature (4 K) scanning tunneling spectroscopy (STS) taken on the monolayer CoCl$_2$ reveals a typical semiconducting band feature with a bandgap -1.7 eV (Fig. 1e), which slightly shift with different moiré modulation (see Supplementary Fig. 15). The Fermi level is close to the edge of the conduction band, suggesting that the

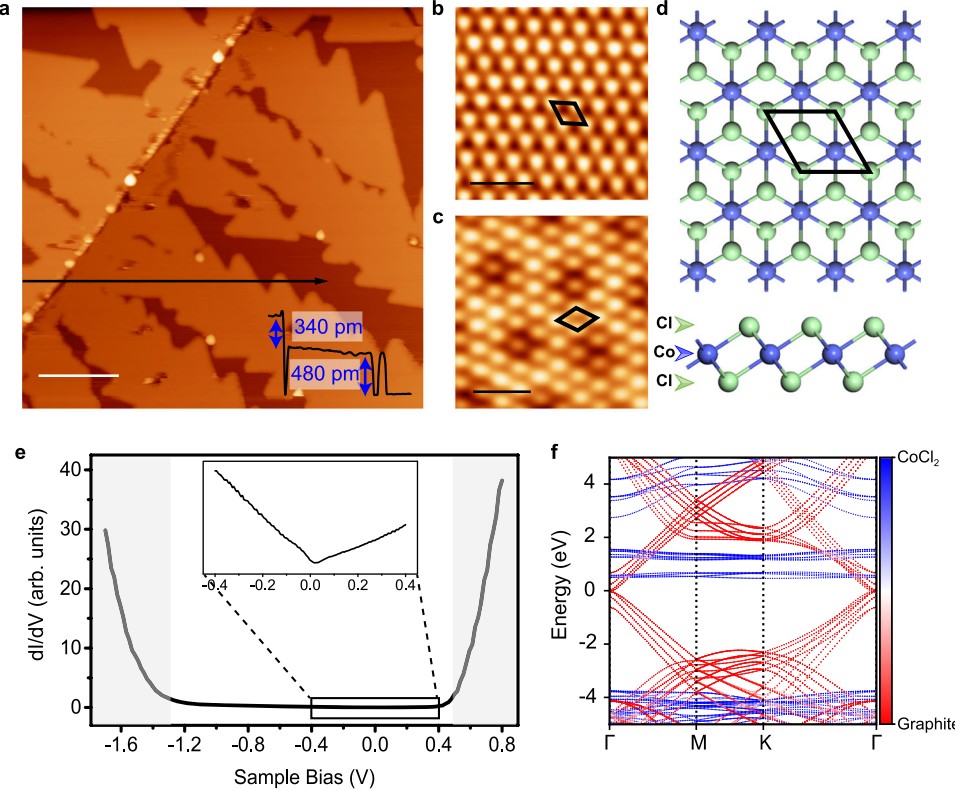

**Fig. 1 | The STM/S characterizations of CoCl$_2$ monolayer on HOPG substrate.**
**a** Large-area STM image ($V_s = -1$ V, $I = 15$ pA) of CoCl$_2$ with sub-monolayer coverage. Inset: line profile along the black arrow across steps of HOPG and CoCl$_2$. Scale bar, 60 nm. **b, c** High-resolution STM images of CoCl$_2$ surface. The protrusions correspond to Cl atoms, **b** exhibits no moiré pattern ($V_s = 750$ mV, $I = 20$ pA) while **c** exhibits moiré periodicity of 1.18 nm ($V_s = -2$ V, $I = 40$ pA) due to different orientations of the CoCl$_2$ monolayer with respect to the HOPG substrate. Scale bars, 1 nm. **d** Top and side views of the atomic structural model of CoCl$_2$ monolayer. Purple and green balls represent Co and Cl atoms, respectively. The 1 × 1 unit cell is labeled by black rhombus in (**b**–**d**). **e** d$I$/d$V$ curve taken on CoCl$_2$ surface revealing the typical semiconducting nature. The inset shows the d$I$/d$V$ curve within small bias range around the Fermi level [−0.4 eV, +0.4 eV]. **f** Calculated band structure of the CoCl$_2$/Graphite by DFT with HSE functional. The red and blue bands are contributed by HOPG and CoCl$_2$, respectively. The reference energy is set to be the Fermi energy.

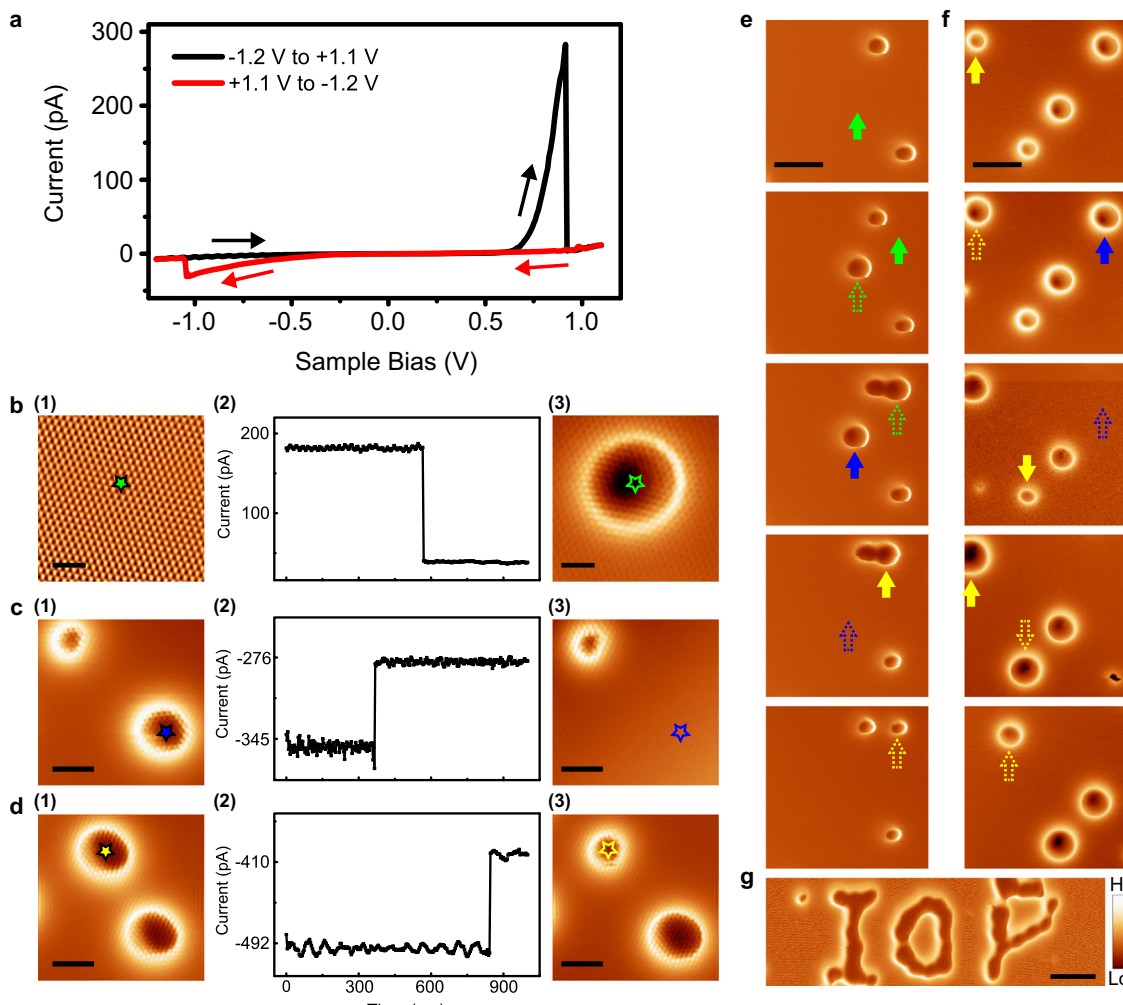

**Fig. 2 | Manipulation process of single polarons. a** Typical *I–V* curve sweeping between −1.2 V to +1.1 V measured on a defect-free region of monolayer CoCl₂ surface, showing two current jumps. **b–d** The process of a single operation event by applying bias pulses. **b** a voltage pulse of 800 mV applying at a clean spot on the surface. **c** A voltage pulse of −1.0 V applying at center of one ring. **d** A voltage pulse of −1.2 V applying at center of the ring. The left panel (**1**) are the d*I*/d*V* maps of an area before event, and the right panel (**3**) are the d*I*/d*V* maps of same area after event. The middle panel (**2**) are the recorded current change during voltage pulsing. The stars mark the tip positions during the pulses. Scale bars, 2 nm for (**b**), and

3 nm for (**c**, **d**). **e**, **f** Two series of d*I*/d*V* maps, from top panels to bottom panels, showing successive manipulation events. The (solid/hollow) green, yellow and blue arrows represent writing, transition and erasure events (before/after), respectively. Scale bars, 9 nm. **g** A d*I*/d*V* map showing an 'IOP' pattern which has been artificially created by successive writing processes. "Hi" and "Lo" in the color scale are the abbreviation of "High" and "Low", corresponding to higher and lower relative intensity of signal. Scale bar, 12 nm. The scanning parameters of all d*I*/d*V* maps are $V_s = 750$ mV, $I = 5$ pA, 90 nm × 23 nm, and with the same tip.

monolayer CoCl₂ is electron-doped. Within the band gap, the d*I*/d*V* curve exhibits a V shape (inset of Fig. 1e) corresponding to the electronic structure of underlying HOPG. For comparison, we calculated the band structure of monolayer CoCl₂ with or without the graphite substrate using density functional theory (DFT). When the graphite substrate is not included in the calculation, the intrinsic band structure of monolayer CoCl₂ has a large bandgap (4.1 eV), shown in Supplementary Fig. 11, which is much larger than the experimental value. While with the inclusion of HOPG substrate the bandgap of CoCl₂/HOPG dramatically decreased to 2.0 eV (Fig. 1f), qualitatively in good agreement with the experiment. As seen in Fig. 1f, the decreased bandgap is mainly due to the lifting of the valence band maximum (VBM) by the states from HOPG, whereas the conduction band is still contributed mainly by the CoCl₂ layer (blue flat band around 0.5 eV). Note that the Dirac cone of HOPG is inside the bandgap, in consistent with the V shape density of states (DOS) in Fig. 1e, but its low DOS seems not affecting the apparent band gap measured in STS. We will come back with more details in the theoretical part.

## Manipulation of single polarons

Interestingly, we found a highly reproducible hysteresis-like phenomenon when performing *I-V* measurements on the CoCl₂ monolayer. Sweeping the sample bias between −1.2 V and +1.1 V, very often we encounter sudden current jumps around ±1.0 V, where the smoothly increasing current suddenly become smaller (Fig. 2a). We first consider the jump at the positive bias side. As positive sample bias corresponds to electron injection from tip to sample, we infer that the injecting electrons may stimulate a critical process when the bias voltage is above a threshold. To show the effect more clearly, we applied a positive bias of 0.8 V with the feedback loop off, and monitored the tunneling current within a time window. Very often a current jump from high current state to a low current state can be recorded, as indicated in Fig. 2b. Consequently, a dramatic ring-like feature with depressed inner region will appear in the d*I*/d*V* map taken afterwards at the same position. Note that in order not to disturb the feature during scanning, we scan the surface with a bias lower than the thresholds.

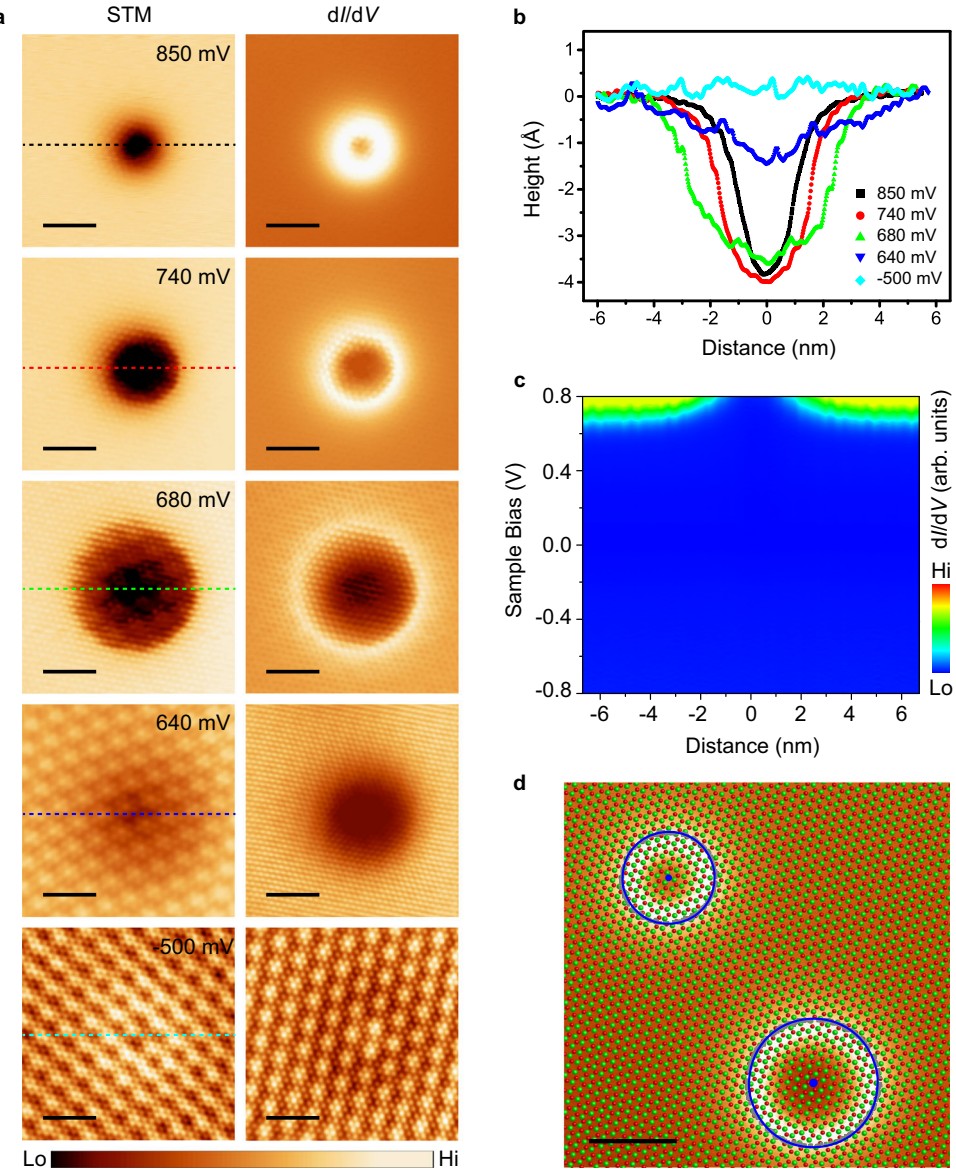

**Fig. 3 | The bias-dependent feature of polaron in STM images. a** Atomic resolution STM images (left panels) and d$I$/d$V$ maps (right panels) of the same individual polaron. The scanning bias from up to down is 850 mV, 740 mV, 680 mV, 640 mV (empty state) and −500 mV (filled state) with tunneling current 25 pA. Scale bars: 3 nm. **b** Height profiles along the lines across the features in (**a**). For a clearer comparison of the change, the height away from the feature is set to be the reference. **c** d$I$/d$V$ map along a line across a ring feature. "Hi" and "Lo" in the color scales are the abbreviation of "High" and "Low". A significant upward bending of conduction band close to the ring center is observed. **d** d$I$/d$V$ maps of monolayer CoCl$_2$ with coexisting type-I (lower right) and type-II (upper left) polarons. Scanning parameter: $V_s$ = 750 mV, $I$ = 10 pA. The atomic lattice of monolayer CoCl$_2$ is superimposed on the image, where cobalt and chloride atoms are red and green balls, respectively. The gray points are raw data extracted from isosurface d$I$/d$V$ = 1.0 pS, which are fitted with circle equation, and locus (blue solid lines) and geometric center (blue points) of the rings can be given. Scale bar, 3 nm.

We then consider the current jump in the negative bias side, and we found that it corresponds to the annihilation of the ring-like feature. As shown in Fig. 2c, when applying a negative bias of −1.0 eV while holding the STM tip on top of an existing ring feature, within a time window very often a current jumping from high current state to a low current state can be recorded. Consequently, we find that the ring-like feature has been annihilated from the surface (panel 3 in Fig. 2c).

Carefully inspecting the ring-like features, we find there are two types of rings, the apparently bigger one and smaller one under the same scanning parameters, labeled as type-I and type-II features, respectively as shown in Fig. 2c, d. Both of them can be created or erased with bias pulse as discussed above. Moreover, the transition between the two types of polarons can also be realized by precisely regulating the voltage pulse. We can also implement all three kinds of

operations sequentially on a single ring. Two examples are shown in Fig. 2e, f, where the green, blue, and yellow arrows represent writing, erasure, and transition operations, respectively. As a demonstration of our capability to manipulate individual ring features, an 'IOP' pattern consisting of 40 rings has been artificially created by successively writing individual rings (Fig. 2g), which means that one can design and implement any artificial patterns with proper manipulation and also suggest the possible application in data storage.

**Evidences of electron polarons**

In order to understand the origin of the ring-like features in monolayer CoCl$_2$, bias-dependent high-resolution STM images have been recorded on a type-I ring we created. As shown in Fig. 3a, the empty sate STM images (positive bias) of same area reveal a smoothly depressed

feature, while the corresponding d$I$/d$V$ maps show a ring-like feature with depressed inner region. In contrast, the filled state STM images and d$I$/d$V$ maps (−500 meV) of same area show a lightly brighter bump, and a lightly darker ring with protruding inner region, respectively. In addition to the dramatic difference of STM images with bias polarity, the bias voltages dependence with the same polarity is also significant. In the empty states, the radii of the features in both the STM images and d$I$/d$V$ maps gradually shrink as the bias increases, as also seen in the height profiles (Fig. 3b) alone the lines marked in Fig. 3a. Such delocalized features with strong bias dependence in STM/STS images usually originates from the charge-induced band-bending by the screened electric field associated with a charged center, such as point defects[43], single dopants[44–47], and adatoms[48,49] on semiconductor surfaces. Therefore, the feature we observed here is very likely caused by a local charge. To further confirm this point, STS was measured along a line across the feature. As shown in Fig. 3c, the conduction band is bended upward around the feature, suggesting that there is a negative charge in the center of the feature.

At this point, one should consider whether this charge center is associated with a surface defect, a sub-surface shallow donor, or a polaron. It is very important that the polaron can be created in arbitrary position set by the STM tip (proven by the artificial "IOP" pattern), meaning that it is impossible to be an existing defect or shallow donor state. In case of existing defects or shallow donors their positions are fixed, and only these specific positions can be charged. Moreover, our high-resolution STM images show perfectly continuous atomic lattice without surface defects before and after the writing process (the bottom panels of Fig. 3a), thus precludes the possibility that a surface defect may be created by the STM tip (for examples an adatom or cluster dropped from the STM tip, or surface atom picked up by the tip to leave a vacancy). The features also be found in bilayer CoCl$_2$ film, as shown in Supplementary Fig. 18, further rules out the contribution from the defect located at the interface of CoCl$_2$/HOPG. Moreover, another important observation in our experiment is the hopping of the features under tip perturbation, often occurring during STM scanning as shown in Supplementary Fig. 12. In order to understand the hopping dynamics of polarons, we have performed first-principles simulations (see Supplementary Fig. 13). The results show the hopping barriers of type-I and type-II polarons are 117 and 72 meV, respectively, which indicate that the polarons can hop easily under tip perturbations. Therefore, in order to obtain a stable STM image of the feature under investigation, the scanning conditions need to be set to minimize the tip perturbation (small bias voltage and tunneling current). Finally, a locally trapped charge in a defect-less lattice, with possible migration, unambiguously coincides with the concept of polaron, which will be further corroborated by our theoretical calculations that will be discussed in the following context.

## Two types of electron polarons

As we mentioned above, we found two types of features with different apparent radius in STM images, named as type-I and type-II. Figure 3d provides more details of the two types of polarons in the atomic resolution d$I$/d$V$ map at 750 meV, clearly showing the ring-like feature with different radius (3.8 nm for type-I and 3.0 nm for type-II). The inclusion of the two types of polarons in the same image allows us to determine their precise registry with respect to the CoCl$_2$ lattice, by superimposing the atomic model of CoCl$_2$ on the d$I$/d$V$ map. We find that the centers of two types of polarons are located at different lattice sites: Type-I polaron is centered on the atop site, whereas that of type-II polaron is centered on the hollow site of the surface Cl layer, as indicated by the blue points. Additionally, in our experiments, type-I polarons are more common than type-II polarons, suggesting that type-I polarons should be slightly more stable than type-II in energy.

It should be emphasized that the size of the depression or ring-like feature (typically a few nanometers) does not represent the size of the

polaron itself which is defined by the size of the distorted region of the lattice. Instead, it corresponds to the spatial extent of the static charge-induced surface band bending. The band-bending modifies the effective tunneling barrier, and produces the dominant features in STM images and d$I$/d$V$ maps. To prove this and to explain the STM images, we performed simulation of the electrostatic potential around the two types of polarons (the method is shown in Part. I of Supplemental Materials), and the results shown in Supplementary Fig. 2 are qualitatively in good agreement with experiments. On the other hand, the structural models obtained by first-principles calculations for both type-I and type-II polarons show only local lattice distortions with sizes comparable to the lattice constant of CoCl$_2$. And thus, both of them fall into the small polaron category, as will be discussed in the next session.

## Theoretical simulation of polarons

To reveal the underlying physical mechanism of polaron formation, DFT simulations were performed with the CP2K/Quickstep method[50,51]. By adding an extra electron into the conduction band minimum (CBM) of monolayer CoCl$_2$ and structure relaxation, we obtained two polarons with or without the initial structure distortions (for more details, see Part. III of Supplementary Materials), and the results are shown in Fig. 4. In Fig. 4a, we plot the spin-polarized projected density of states (PDOS) of first polaron. With some structure distortion, a localized polaron state emerges in the band gap (labeled as Polaron I), which is at 0.49 eV below the CBM and contributed by Co_$3d$ orbitals. For second polaron (labeled as Polaron II), shown in Fig. 4c, the charge trapping by lattice distortion affects the electronic structure more distinctly. Two polaron-induced peaks are formed from 0.31 to 0.64 eV below the CBM. Figure 4b, d give the lattice distortions and charge distribution of CoCl$_2$ for the two types of polarons, and we found the formation of both polarons still preserves the triplet symmetry of system. From the top view, polaron I locates on three Co atoms forming a triangle. The $d$ orbitals of the three Co atoms hybridize with each other and form a bonding orbital, locating in the center of the triangle, and pushes the three Co atoms to the central direction by 0.15–0.22 Å. Then the center localized charge pushes the central Cl atom downward by 0.20 Å, which can be clearly seen from the side view. The three sublayer Cl atoms also move upward by 0.04 Å. More details related to the bond length change can be seen in Supplementary Materials part. III. For polaron II, shown in Fig. 4d, the additional charge locates on one Co atom. And it pushes the six Cl atoms away by less than 0.10 Å. Polaron II is consistent with the property of general electron polarons that it repels surrounding anions, and polaron I shows some difference because the charge locates on three Co atoms and there is orbital hybridization between them, which plays a role in the lattice distortion. According to the PDOS analysis (see Fig. 4a, c), the charge state on the three central Co atoms of polaron I changes from +2.0 to +1.7 e, which is consistent with the excess electron being delocalized on three Co atoms and each of them acquires 0.3 electron. In contrast, for polaron II, the charge state changes from +2.0 to +1.2 e, consistent with a full localization of the excess electron in one Co atom. For polaron I and II, the lattice distortions occur within several unit cells and the range radii are around 5.40 Å and 3.50 Å, respectively, which match to typical small polaron-induced lattice distortions. Note that such small lattice displacement is beyond the spatial resolution limit of STM. For polaron I, the charge density is strongly localized on the center of three Co atoms, whereas the polaron II has the charge concentrated on a single Co atom. These features are in good agreement with the experimental observation of two types of polarons located at atop site and hollow site on the surface. So, we confirm the Polaron I and II are type-I and type-II polarons, respectively.

As we discussed before, the dominant ring-with-depression feature in STM images is due to the electric field- induced band bending. The broader electron distribution of type-I polaron in three Co atoms

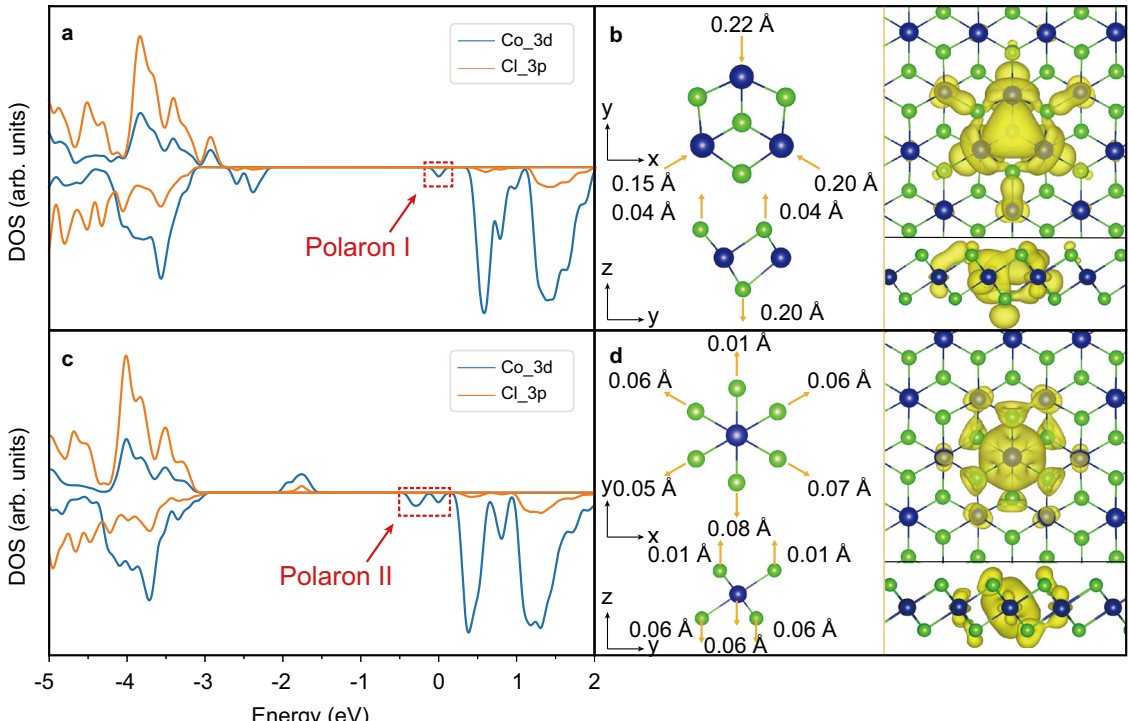

**Fig. 4 | Modeling two types of polarons in monolayer CoCl2 based on theoretical simulations.** The up (**a**, **b**) and down panels (**c**, **d**) correspond to calculations of type-I and type-II polarons, respectively. **a**, **c** Spin-polarized density of states (DOS) of monolayer CoCl$_2$ with single polarons. The upper and lower parts represent spin up and down, respectively. The reference energy is VBM energy. The red dotted rectangle marks the in-gap state of polaron. **b**, **d** Left panels: Structure distortion diagrams of monolayer CoCl$_2$ induced by two types of polarons. The arrows represent the directions of atom distortion. Purple and green balls represent cobalt and chlorine atoms, respectively. Right panels: The charge density distribution (up panel: top view, down panel: side view) of two types of polarons.

gives slightly broader band-bending, and thus well explains the observed bigger ring-like feature in type-I polaron as compared with type-II polaron (more details in Supplementary Materials part. I). The binding energies ($E_p$) for two types of polarons are 0.31 eV and 0.22 eV, respectively, indicating the divergent type-I polarons are more stable in energy than the concentrated type-II, which is in line with the experimental statistics. Considering the influence of substrate, we have performed the calculation with or without the HOPG substrate. The results with HOPG are shown in Supplementary Fig. 10 in Supplementary Information. Both type-I and type-II polarons can be maintained with or without HOPG, so we suggest that the graphite substrate does not change the major polaron nature.

Moreover, one should have noticed that the in-gap polaron states indicated by calculation were not be observed on STS measurement. One possible reason would be that the DOS of these states are too weak and thus they are hidden in the background noise and/or substrate signal in STS (see Supplementary Fig. 17). Another possible, and more interesting reason is that most previous experiments deal with the small polarons bound with defects, while we deal with intrinsic polarons in a defect- less lattice. Hence, compared with the sharp polarons peaks reported previously, the polaron states of our free polarons could be much weaker and could be easily masked by background signals.

## Mechanism of the polarons manipulation

To find out the mechanism of polaron manipulation, we statistically analyzed the writing probability as a function of the pulse voltage, as shown in Fig. 5a. In our experiments, the writing probability is defined as ratio of the number of writing events to the number of applied pulses (>100 times) with same energy, current and duration time). When the energy of tunneling electrons is low enough, the writing probability remains almost zero. With the increase of pulse voltage

above 800 mV, the ratio of writing events rapidly increases to one, which means nearly 100% possibility of writing for every bias pulsing when electron energy exceeds the threshold value. The Boltzmann fitting gives a threshold voltage about 858 meV at a tunneling junction set by ($V_s$ = 600 mV, $I$ = 25 pA).

The writing process is directly related to the injection of electron to the conduction band of CoCl$_2$ monolayer on HOPG. We calculated the DOS of CoCl$_2$ monolayer involving the HOPG substrate. As shown in Fig. 5b, the CoCl$_2$ monolayer keeps its insulating property, while the Fermi surface of graphite is located inside the gap of CoCl$_2$. The total DOS near Fermi level shows a V shape, and it starts to deviate from the Dirac cone around −1.5 eV and +0.5 eV, which is well in agreement with our experimental results (Fig. 1e). The calculated energy position of conductive band edge of CoCl$_2$ (520 meV) is smaller than the experimental writing threshold energy (858 meV). The larger threshold voltage in experiment may be due to the formation of the double barrier in the tunneling junction: one is between tip and the surface of CoCl$_2$ monolayer, and the other between the upper and lower surface of CoCl$_2$ monolayer. The actual bias voltage dropped between tip and CoCl$_2$ monolayer is thus smaller than the bias voltage applied to the tunneling junction.

Finally, it is worth mentioning the role of HOPG in the formation of polarons. As control experiment, we have performed the same experiments using Au(111) substrate. We found that although perfectly crystalline CoCl$_2$ can be obtained on Au(111), there is no sign of polaron formation in this case (experimental dada shown in Supplementary Fig. 19 in Supplementary Materials Part. V). Therefore, HOPG may play important roles in the formation of polarons. On the one hand, the conducting nature of HOPG substrate allows STM and polaron manipulation experiments to be carried out. On the other hand, the low carrier density and weak interaction between CoCl$_2$ layer and the substrate avoids the fast compensation of residual charge in the CoCl$_2$ layer.

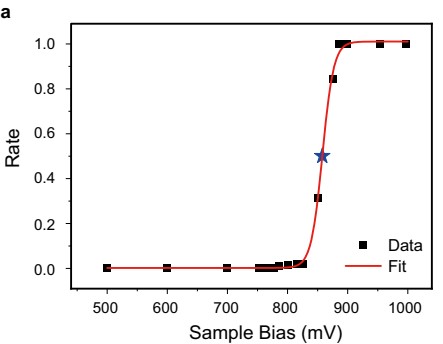

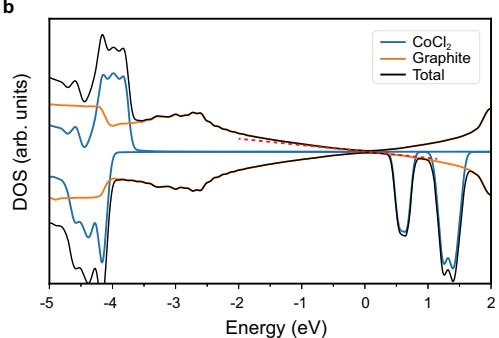

**Fig. 5 | The manipulation mechanism of polaron. a** The statistics of writing rate of polaron with sample bias. The rate is defined as the ratio of the number of successful writes to the total number of operations with the same condition. The solid red curve is the result of Boltzmann fitting, which gives the threshold bias of 858 meV with pulse width of 1 s at tunneling parameter $V_s = 600$ mV, $I = 25$ pA. **b** The

DOS of CoCl₂ monolayer on four-layer graphite. The upper and lower parts represent spin up and down, respectively. The red dotted line represents the linear DOS of Dirac cone. The reference energy is Fermi energy. The edge of CoCl₂ conduction band is about 520 meV above Fermi level, which is close to the experimental threshold bias of writing.

## Discussion

In conclusion, we discovered electron polarons in a two-dimensional semiconductor, CoCl₂ monolayer on HOPG substrate. Systematic investigations were performed to characterize the polarons, including the location of the lattice distortion, charge distribution and band bending. Remarkably, for the first time, single polaron can be created and manipulated at atomic level with the STM tip. Note that controllable charge states have only been realized in surface defect or adatoms where the binding energies are much larger. Such controllable, intrinsic polarons in defect-less CoCl₂ monolayer provide us an ideal system for both the study of polaron physics in atomic scale. Furthermore, each polaron should be a single spin carrier, and its interaction with the magnetism of CoCl₂ film may be more fascinating, potential for more study and applications in data storage, spintronics and quantum computing.

During the submission process of our manuscript, it came to our attention that a parallel study by Cai et al. was uploaded[52], also reported the observation of electron polarons and gave a simple description of their manipulation in the same material system. Notably, they presented four distinct types of polarons. The type-1 and type −2 polarons in their study are most likely to correspond to type-I and type-II polarons in this work. However, Cai et al. propose that both types result from the trapping of a single electron on a single Co atom, without the involvement of phonon coupling. In contrast, our type-I and type-II polarons result from electron trapping on three adjacent Co atoms and a single Co atom with associated lattice distortions, respectively. Our experiments did not observe the type-3 and type-4 polarons reported by Cai et al. This could be attributed to differences in experimental temperature or possible local defects induced by STM tip in their experiment.

## Methods

### Sample fabrication and STM/S characterization

Our experiments were carried out in a home-built low-temperature STM/MBE system with a base pressure better than $5 \times 10^{-11}$ Torr. A clean HOPG substrate was prepared by mechanical exfoliation in air, and annealing at 1000 K in ultrahigh vacuum. The CoCl₂ monolayer were prepared by directly evaporating anhydrous CoCl₂ beads (Sigma, 99.999%) on HOPG substrate kept at room temperature. After growth, the sample was transferred to the STM chamber and measured with a tungsten tip at 4 K. The STS were measured by a lock-in technique, in which an ac voltage of 20 mV and 659 Hz was superimposed on the given sample bias. The STM images and d$I$/d$V$ maps were all obtained at constant-current mode if not specified.

### Theoretical calculation based on DFT

The geometry optimization and electronic structure of polarons were performed with the quickstep module of the CP2K program package[50,51] within the Gaussian and Plane Waves (GPW) framework. HSE06[53,54] exchange-correlation functional together with Goedecker–Teter–Hutter (GTH) pseudopoetentials[55] was applied. The cutoff and relative cutoff energies of the auxiliary plane wave basis sets were converged to energy differences smaller than $10^{-6}$ hartree/atom. Triple-zeta "MOLOPT" basis sets[56] were used for Co and Cl. The CoCl₂ monolayer was modeled using a hexagonal 6 × 6 supercell with 108 atoms sampled at the Γ point. A vacuum space larger than 25 Å was adopted to avoid any interaction between two adjacent slabs.

The band structure and DOS of CoCl₂ monolayer and CoCl₂/HOPG were preformed using the Fritz-Haber-Institute ab initio molecular simulations (FHI-aims) package[57,58]. A scalar relativistic treatment with the atomic ZORA approximation[59] was included in calculations. We used the "light" setting for numerical atom-centered orbital basis sets in FHI-aims. In our calculations, the CoCl₂/HOPG system was built by depositing a 2 × 2 supercell of monolayer CoCl₂ on a 3 × 3 supercell of 4 layers' graphite. The Brillouin zone was sampled by a (48 × 48 × 1) k-point mesh. The lattice mismatch between the CoCl₂ and graphite substrate in the constructed heterostructure was about 4%. We have compared the band structure of free-standing CoCl₂ with and without strain and their difference is negligible (for more detail, see Part. III in the Supplementary Information). The vdW interaction functional using the method of Tkatchenko-Scheffler method with iterative Hirshfeld partitioning[60] was employed in the vdW heterostructure calculations. The structures were relaxed using a Broyden–Fletcher–Goldfarb–Shanno (BFGS) optimization algorithm until the maximum force on each atom was less than 0.01 eV Å⁻¹. The convergence criteria of $10^{-6}$ eV for the total energy of the systems were used.

## Data availability

All data that support the findings of this paper are available from the corresponding authors upon request.

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

## Acknowledgements

This work was financially supported by the Ministry of Science and Technology (MOST) of China (2021YFA1400500, 2018YFE0202700, 2021YFA1202902), National Natural Science Foundation of China (11825405, 1192780039, 12134019, 11974322, 12125408), the Chinese Academy of Science (XDB30000000, YSBR-054). Calculations were performed Hefei Advanced Computing Center, ORISE supercomputing center and Supercomputing Center at USTC.

## Author contributions

K.W. and L.C. designed supervised the project. H.L. performed experiments and data analysis. A.W. perform first-principles calculations under the supervision of J.Z. H.L. and K.W. prepared the manuscript with contributions from L.C., A.W., and J.Z. All other authors contributed to the experimental setup and discussion during the research in this project.

## Competing interests

The authors declare no competing interests.
