## [Peer Review File · Nature Communications]

Reviewers' Comments:

Reviewer #1:

Remarks to the Author:

I have previously reviewed the paper "Atomic-scale Manipulation of Single-Polaron in a Two-Dimensional Semiconductor" for Nature Materials (August 2022). Now the paper has been submitted to Nature Communications. In my original report (Referee 3 of Nature Materials paper) I raised a number of questions/comments, but overall I expressed a positive opinion on the paper. In their revised version (September 2022) the authors provided extensive answers to the comments of the referees, including mine. The paper was definitely improved by these changes. I have read once more the present version of the paper, submitted to Nature Communications, and I confirm my overall positive evaluation. There is only one minor point where things could be further improved.

In their answer to my comment 6, the authors mentioned the change in charge state of the Co ions as determined by the atomic charges from PDOS analysis. In particular, the charge of Co in type-I polaron is changing from +2.0 to +1.7 e, which is consistent with the excess electron being delocalized on three Co ions. For type-II polaron the change in charge goes from +2.0 to +1.2 e, consistent with a full localization. I think these data are useful, and could be mentioned in the main text.

For the rest the paper can be published in present form.

Reviewer #2:

Remarks to the Author:

Referee Report on manuscript "Atomic-scale Manipulation of Single-Polaron in a Two-dimensional Semiconductor" by Liu H. et al. (manuscript no.: NCOMMS-23-11498)

The manuscript by Liu et al. reports a STM/S study of single polarons in a monolayer of transition metal dihalide CoCl_2 , itself grown on a HOPG substrate by means of molecular beam epitaxy. Using voltage pulses in STM, Liu et al. demonstrated atomic-scale manipulation, including creation, erasure, displacement, and inter-conversion of polarons within the CoCl_2 lattice, which the authors further classified into two different types based on their appearances, site occupations and spectroscopic signatures in STM. As control, the authors also performed a similar experiment using a Au(111) single crystal as a substrate, where they found no sign of polaron formation, highlighting the important role of the HOPG substrate in the polaron formation within the CoCl_2 layer.

Numerous works have attempted to study the properties of single polarons, with most focusing on polarons induced by dopants, or by defects formed within the bulk or at the surface. However, experimental studies of single polarons formed within a perfect, defect-free lattice are rare. Here using $\text{CoCl}_2/\text{HOPG}$ as the material system of study, Liu et al. demonstrated manipulation as well as characterization of single polarons with atomic-scale precision. Their work is both scientifically interesting and important, and should be able to attract wide audiences. Addressing the following questions/comments are required before warranting its publication as a Research Article in Nature Communications:

1. Another work by Cai et al. (arXiv:2207.04948) shows using the same material system pulse-induced formation of four different types of polarons, rather than only two as reported by this manuscript. One possible reason is that the two experiments were carried out at very different temperatures. Could the authors comment on this in the main text? In addition to temperature, is there any other factors that can also cause such difference?
2. The calculated appearances of the polarons are in good agreement with the experimental counterparts. However, no in-gap states were observed in the experiments, which seems to contradict the predictions from the calculations, and intuitive thinking for polaron formation. Is it possible that the pulse-induced protrusions observed by Liu et al. are indeed electrons trapped around the Co^{4+} ions within the CoCl_2 lattice, without Co^{4+} being turned into Co^{3+} (in other words, single charge trapping with no polaron formation)? Can DFT calculations confirm this?
3. It is generally believed that at sufficiently high temperatures self-trapped polarons can hop between different sites within the lattice. To demonstrate polaron hopping, one can either perform similar experiments at different temperatures to see if polarons move, or perform at low

temperature (4 K) action spectroscopy to determine the related kinetics parameters. Since being capable of demonstrating polaron hopping and understanding the related properties are very important to polaronic research, could the authors also provide the relevant data (if they have), or at least comment on this point in the main text?

Reviewer #3:

Remarks to the Author:

I have already reviewed this paper for Nature Materials. I have read the latest version of the manuscript and compared it with the parent paper by Cai and coworkers, 'Manipulating single excess electrons in monolayer transition metal dihalide'.

I confirm that the authors have done excellent work in addressing all issues raised in the first round of review but did not address the last points that I pointed out in my previous NatMat report when submitting their work in NCOMMS.

In particular, the authors have not included either a discussion or a reference to the paper of Cai and coworkers available on arxiv.org/abs/2207.04948.

I recommend publishing this paper in Nature Communications only after the author includes a brief comparative comment of their results with those of Liu in the main text and adds a reference to Liu's paper.

Reply to referee #1:

C: *I have previously reviewed the paper "Atomic-scale Manipulation of Single-Polaron in a Two-Dimensional Semiconductor" for Nature Materials (August 2022). Now the paper has been submitted to Nature Communications. In my original report (Referee 3 of Nature Materials paper) I raised a number of questions/comments, but overall I expressed a positive opinion on the paper. In their revised version (September 2022) the authors provided extensive answers to the comments of the referees, including mine. The paper was definitely improved by these changes. I have read once more the present version of the paper, submitted to Nature Communications, and I confirm my overall positive evaluation. There is only one minor point where things could be further improved.*

R: Thanks for the referee's positive evaluation and constructive comments.

C1: *In their answer to my comment 6, the authors mentioned the change in charge state of the Co ions as determined by the atomic charges from PDOS analysis. In particular, the charge of Co in type-I polaron is changing from +2.0 to +1.7 e, which is consistent with the excess electron being delocalized on three Co ions. For type-II polaron the change in charge goes from +2.0 to +1.2 e, consistent with a full localization. I think these data are useful, and could be mentioned in the main text. For the rest the paper can be published in present form.*

R1: Thanks for the suggestion.

We added the following discussion to the main text (Page 13): "According to the PDOS analysis (see Figs. 4a and 4c), the charge state on the three central Co atoms of polaron I changes from +2.0 to +1.7 e, which is consistent with the excess electron being delocalized on three Co atoms and each of them acquires 0.3 electron. In contrast, for polaron II, the charge state changes from +2.0 to +1.2 e, consistent with a full localization of the excess electron in one Co atom."

Reply to referee #2:

C: *Referee Report on manuscript "Atomic-scale Manipulation of Single-Polaron in a Two-dimensional Semiconductor" by Liu H. et al. (manuscript no.: NCOMMS-23-11498)*

The manuscript by Liu et al. reports a STM/S study of single polarons in a monolayer of transition metal dihalide CoCl₂, itself grown on a HOPG substrate by means of molecular beam epitaxy. Using voltage pulses in STM, Liu et al. demonstrated atomic-scale manipulation, including creation, erasure, displacement, and inter-conversion of polarons within the CoCl₂ lattice, which the authors further classified into two different types based on their appearances, site occupations and spectroscopic signatures in STM. As control, the authors also performed a similar experiment using a Au(111) single crystal as a substrate, where they found no sign of polaron formation, highlighting the important role of the HOPG substrate in the polaron formation within the CoCl₂ layer.

Numerous works have attempted to study the properties of single polarons, with most focusing on polarons induced by dopants, or by defects formed within the bulk or at the surface. However, experimental studies of single polarons formed within a perfect, defect-free lattice are rare. Here using CoCl₂/HOPG as the material system of study, Liu et al. demonstrated manipulation as well as characterization of single polarons with atomic-scale precision. Their work is both scientifically interesting and important, and should be able

to attract wide audiences. Addressing the following questions/comments are required before warranting its publication as a Research Article in Nature Communications:

R: Thanks for the referee's constructive comments. We have addressed all the questions point-by-point in the following.

C1: *1. Another work by Cai et al. (arXiv:2207.04948) shows using the same material system pulse-induced formation of four different types of polarons, rather than only two as reported by this manuscript. One possible reason is that the two experiments were carried out at very different temperatures. Could the authors comment on this in the main text? In addition to temperature, is there any other factors that can also cause such difference?*

R1: Thanks for the suggestions.

The work uploaded to arXiv by Min Cai et. al. (arXiv: 2207. 04948), which is two months later after our work upload on arXiv (arXiv: 2205. 10731), also reported the observation of electron polarons and give a simple description of their manipulation. In their manuscript, they showed four different types of polarons. The type-1 and type-2 polarons in their work are mostly likely the type-I and type-II polarons described in our manuscript. However, Cai et al. suggest that the type-1 and type-2 polarons are both from one electron trapped on single Co atom, without the contribution of phonon coupling, while our type-I and type-II polarons in our work are described as one electron trapped in three adjacent Co atoms and single Co atoms accompanied by lattice distortions, respectively.

The type-3 polaron Cai's work shows a depression that is indistinguishable with the type-1 polaron, but a protrusion in higher voltage, as well as a similar energy shift of conduction band edge. The type-4 polaron in their work only be observed at liquid nitrogen temperature. Both types of polarons have not be theoretically explained by them. Since these two types of polarons are not observed in our experiments, we think such difference could be due to tip-induced local defects or additional feature related with the temperature. Further investigation would be required in order to clarify this issue, but it is out of the scope of this manuscript.

For a brief comparison of the two works, in our revised manuscript, the following statements are added (Page 25): "During the submission process of our manuscript, it came to our attention that a parallel study by Cai et al. was uploaded (arXiv: 2207. 04948), also reported the observation of electron polarons and gave a simple description of their manipulation in the same material system. Notably, they presented four distinct types of polarons. The type-1 and type-2 polarons in their study are most likely to correspond to type-I and type-II polarons in this work. However, Cai et al. propose that both types result from the trapping of a single electron on a single Co atom, without the involvement of phonon coupling. In contrast, our type-I and type-II polarons result from electron trapping on three adjacent Co atoms and a single Co atom with associated lattice distortions, respectively. Our experiments did not observe the type-3 and type-4 polarons reported by Cai et al. This could be attributed to differences in experimental temperature or possible local defects induced by STM tip in their experiments."

C2: *2. The calculated appearances of the polarons are in good agreement with the experimental counterparts. However, no in-gap states were observed in the experiments, which seems to contradict the predictions from the calculations, and intuitive thinking for polaron formation. Is it possible that the pulse-induced protrusions observed by Liu et al. are indeed electrons trapped around the Co⁴⁺ ions within the CoCl₂ lattice, without Co⁴⁺ being turned into Co³⁺ (in other words, single charge trapping with no polaron formation)? Can DFT calculations confirm this?*

R2: Thanks for the referee's comment.

In our experiments, some signal about in-gap states are indeed observed, but it's extremely weak, and requires a close tip-sample distance. As shown in **Fig. R1**, with the decreasing of tip-sample spacing (tunneling current set from 5 pA to 25 pA), some weak in-gap states appear in the background of the band bending. Figure R1d show the contrast of the dI/dV spectrum obtained at the center of the polaron (black curve) and away from the polaron (red curve) when the tunneling current up to be 1 nA. And these data have been added into the revised Supplementary Information.

Fig. R1 The in-gap features of the dI/dV spectra. (a-c) Line color maps across one polaron with the increased tunneling current, corresponding to 5 pA, 10 pA and 25 pA. (d) dI/dV curves with tunneling current of 1 nA. Black and red curves correspond to the spectrum at the center of the polaron and away from the polaron.

On the other hand, our theoretical results shows that the localized charge densities of type-I and -II polarons cannot be stably sustained in the absence of structural distortions. As shown in **Fig. R2**, we introduced an electron into the undistorted structure, followed by an SCF calculation. It can be seen that the charge distribution is relatively diffuse and irregular, which is different with the charge distribution of type-I and -II polarons and is inconsistent with the STM map. Therefore, we believe the type-I and -II polarons in our experiments are the conventional small polarons under electron-phonon coupling.

Fig. R2 The charge distribution of excess electron in undistorted structure.

C3: 3. *It is generally believed that at sufficiently high temperatures self-trapped polarons can hop between different sites within the lattice. To demonstrate polaron hopping, one can either perform similar*

experiments at different temperatures to see if polarons move, or perform at low temperature (4 K) action spectroscopy to determine the related kinetics parameters. Since being capable of demonstrating polaron hopping and understanding the related properties are very important to polaronic research, could the authors also provide the relevant data (if they have), or at least comment on this point in the main text?

R3: Thanks for referee’s comment and suggestion.

We agree that performing statistical measurement on the hopping kinetics of polarons and its dependence on temperature would be interesting and important to understand the physics of polarons. We have indeed tried to perform such experiment (see Supplementary Fig. 12). However, it seems that the tip may induce strong perturbation to the polaron hopping. This is because polaron is a charged object and it strongly coupled with an external field, resulting in such experimental data unreliable.

Alternatively, we have performed first-principles simulations and use a linear interpolation method to calculate the hopping barriers between two type-I polarons and two type-II polarons. There are 20 interpolation points used between the initial and final structures. We also calculate the transition barrier between type-I and –II polarons, and the results are shown in **Fig. R3**. The hopping barrier of type-I and -II polarons are 117 and 72 meV, respectively, and the transition barrier between type-I and type-II polarons is as small as 38 meV. Since linear interpolation method tends to overestimate the energy barriers, the real hopping and transition between these polarons are likely smaller, suggesting that the polaron hopping and transition events can easily occur at room temperature or under perturbations.

We have added the following statement in the revised manuscript (Page 9): “In order to understand the hopping dynamics of polarons, we have performed first-principles simulations (see Supplementary Fig. 13). The results show the hopping barriers of type-I and type-II polarons are 117 and 72 meV, respectively, which indicate that the polarons can hop easily under tip perturbations.”

Fig. R3 Hopping barriers between two type-I (a), type-II (b) polarons, and transition barrier between type-I and type-II polarons (c). The insets show the charge distribution of the initial and final polaron

states. There are 20 interpolation points between the initial and final structures.

Reply to referee #3:

C: *I have already reviewed this paper for Nature Materials. I have read the latest version of the manuscript and compared it with the parent paper by Cai and coworkers, 'Manipulating single excess electrons in monolayer transition metal dihalide'. I confirm that the authors have done excellent work in addressing all issues raised in the first round of review but did not address the last points that I pointed out in my previous NatMat report when submitting their work in NCOMMS. In particular, the authors have not included either a discussion or a reference to the paper of Cai and coworkers available on arxiv.org/abs/2207.04948. I recommend publishing this paper in Nature Communications only after the author includes a brief comparative comment of their results with those of Liu in the main text and adds a reference to Liu's paper.*

R: Thanks for the recommendation and suggestion.

We added a brief comparative comment of our results with the work by Cai et al. in revised manuscript, which is also cited as a reference: “During the submission process of our manuscript, it came to our attention that a parallel study by Cai et al. was uploaded (arXiv: 2207. 04948), also reported the observation of electron polarons and gave a simple description of their manipulation in the same material system. Notably, they presented four distinct types of polarons. The type-1 and type-2 polarons in their study are most likely to correspond to type-I and type-II polarons in this work. However, Cai et al. propose that both types result from the trapping of a single electron on a single Co atom, without the involvement of phonon coupling. In contrast, our type-I and type-II polarons result from electron trapping on three adjacent Co atoms and a single Co atom with associated lattice distortions, respectively. Our experiments did not observe the type-3 and type-4 polarons reported by Cai et al. This could be attributed to differences in experimental temperature or possible local defects induced by STM tip in their experiment.”